# The effects of Arabian jasmine on zebrafish behavior depends on strain, sex, and personality

**Tripatchara Atiratana[1,2], Aliyah R. Goldson[2], Siritron Samosorn[3], Neha Rajput[2], Nalena Praphairaksit[1], Justin W. Kenney[2]\***

**1** Department of Biology, Faculty of Science, Srinakharinwirot University, Bangkok, Thailand,
**2** Department of Biological Sciences, Wayne State University, Detroit, Michigan, United States of America,
**3** Department of Chemistry, Faculty of Science, Srinakharinwirot University, Bangkok, Thailand

\* jkenney9@wayne.edu

## Abstract

*Jasminum sambac* (L.) Aiton, commonly known as Arabian jasmine, is widely used in Thai traditional medicine for mental health ailments. While most studies in humans and animals find that Arabian jasmine reduces stress and anxiety, there are a handful of reports that it can oppose relaxation by increasing autonomic arousal. Using adult zebrafish, we sought to determine whether factors like strain, sex, and personality might contribute to the variable effects of *J. sambac* on anxiety-related behavior. We extracted the flowers of *J. sambac* by ultrasonic-assisted extraction with optimal air pressure. Headspace solid-phase microextraction with gas chromatography-mass spectrometry (HS-SPME-GC-MS) identified the main components in the Arabian jasmine flower extract, including linalool (an anxiolytic compound) and benzaldehyde (a potentially anxiogenic compound). We fed three strains of zebrafish (AB, TL, and WIK) a gelatin pellet containing different concentrations of *J. sambac* (5−20 mg kg$^{-1}$) and assessed 3-dimensional swim behavior in the novel tank and mirror biting tests. We found that in female AB fish, *J. sambac* resulted in a decrease in both bottom distance and percent explored during the novel tank test, consistent with an anxiogenic effect; there was no effect in WIK or TL fish. We also found that behavior/personality type influenced the effects of *J. sambac* where shy AB females increased their percent explored and low activity males increased their bottom distance, consistent with anxiolytic effects. Thus, we find that sex, genetics, and personality interact to influence the anxiety-related effects of Arabian jasmine. This suggests that these factors may contribute to the opposing effects of jasmine previously reported in the literature.

## Introduction

*Jasminum sambac* (L.) Aiton (common name "Arabian jasmine", Thai name "Ma-li-la") is a plant that is commonly used in the traditional medicine of many countries. For example, in Thailand, this plant species is contained in the Ya Hom Thep Pa Chit preparation, which is

**Data availability statement:** All underlying data for the study is available from the figshare repository (https://figshare.com/s/e1a6fa2ff4a5e815808b).

**Funding:** National Institutes of General Medical Sciences (NIGMS; R35GM142566 to JWK) Science Achievement Scholarship of Thailand (to TA and NP). The funders had no role in study design, data collection and analysis, decision to publish, or preparation of the manuscript.

**Competing interests:** The authors have declared that they have no competing interests.

made of 50% Arabian jasmine flowers and consumed orally as an infusion to reduce anxiety [1]. Consistent with an anxiolytic effect, Arabian jasmine has been found to result in an increase in self-reported levels of relaxation and reduced levels of stress hormones [2,3]. However, there are also reports that Arabian jasmine can stimulate arousal [4,5], which would oppose relaxation. Thus, it is unclear how Arabian jasmine can have these two opposing properties. It may be that biological factors of participants, like sex, background genetics, and personality, may influence an individual's responses to Arabian jasmine.

Arabian jasmine is rich in phytochemicals, with benzaldehyde and linalool being particularly prominent [6,7]. Linalool, which has a floral or spicy-wood scent, has been found to reduce anxiety-related behaviors and reduce aggression [8–10]. Benzaldehyde is characterized by an almond-like odor, and although its behavioral effects have not been as well studied, there is at least one report of it increasing aggression [11]. These compounds, alongside several others, create a complex chemical composition that is affected by specific cultivation practices and could contribute to some of the conflicting effects of Arabian jasmine.

Individuals can differ in their behavioral and physiological responses to compounds due to variation in factors such as sex, genetics, and even personality traits. This differential sensitivity provides a potential explanation for paradoxical drug effects (i.e., outcomes opposite to those expected) observed with some drugs. For example, sex can influence how animals respond to drugs like alcohol [12,13]. Genetic variation has long been known to modulate the effects of psychoactive drugs in both model organisms and humans, giving rise to the field of pharmacogenomics [14]. Finally, personality-like traits have also been found to influence the behavioral effects of drugs [15,16]. Thus, it may be the case that the paradoxical effects of Arabian Jasmine that have been reported are due to variation in genetic background, sex, or personality traits of subjects. To test this idea, we examined how genetic background, sex, and personality influences how zebrafish respond to the effects of Arabian jasmine during exploration of a novel environment.

Zebrafish [*Danio rerio* (Hamilton,1822)] are a tropical freshwater fish native to south Asia [17] that has grown in popularity as a model organism in behavioral research over the past several decades [18–20]. This popularity is due to their small size and affordability coupled with high genetic similarity to humans [21]. Zebrafish also possess all the main neurotransmitter systems as mammals [22] with overlapping molecular mechanisms [23], and stress hormones [24]. Finally, adult zebrafish exhibit a wide variety of behaviors [25] making them a viable animal model for behavioral and pharmacological studies. With respect to sex, genetics, and personality, recent studies have found that, like mammals, zebrafish behavior is influenced by these factors [26,27], and that they display consistent individual differences in behavior (i.e., personality) [27,28].

## Results

### Chemical profile of Arabian jasmine

To obtain an extract of Arabian jasmine (Fig 1A), we used ultrasonication with air pressure and water as a solvent (Fig 1B). The aqueous extract was then analyzed

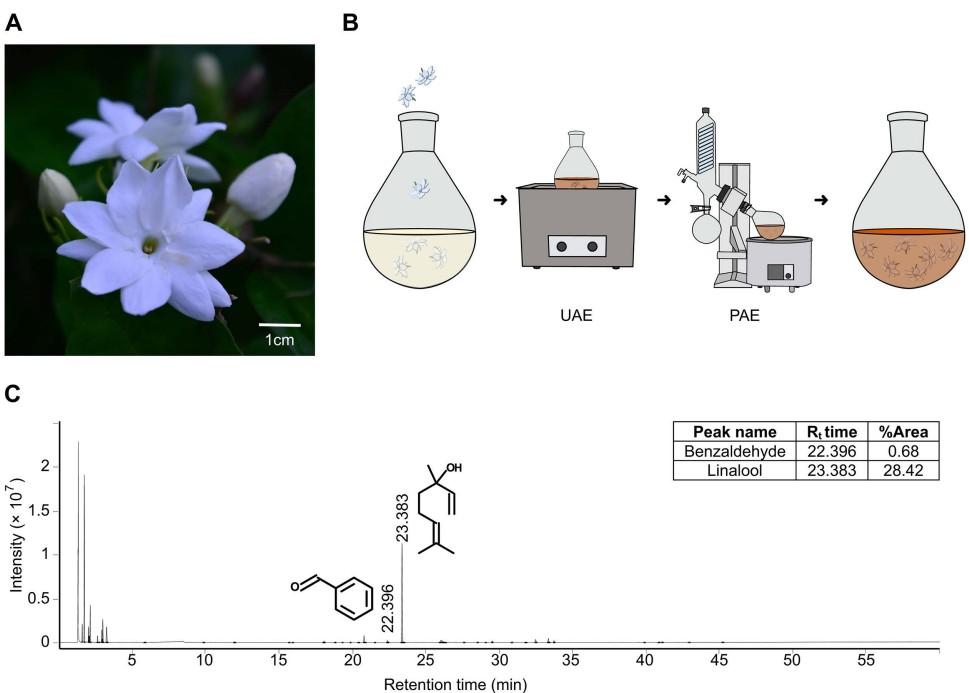

**Fig 1. The chemical studies of Arabian jasmine flower. (A)** Arabian jasmine flowers. **(B)** Experimental design for extracting Arabian jasmine flowers mimicking a traditional approach. These flowers were extracted in water by ultrasonic-assisted extraction (UAE) followed by pressure-assisted extraction (PAE). **(C)** GC chromatogram with MS spectra of Arabian jasmine showing linalool and benzaldehyde. Retention time ($R_t$ time) is the amount of time a compound spends in the GC column before being detected by MS. The % area indicates the relative amount of each compound based on the size of its peak in the GC chromatogram.

via Headspace Solid-Phase Microextraction coupled with Gas Chromatography–Mass Spectrometry (HS-SPME-GC-MS) (Fig 1C and S1 Table). Among the 40 molecules identified, the top two comprised over half the sample (dimethyl sulfide: 30.2% and linalool: 28.4%). Other compounds were detected at lower concentrations: 3-hexen-1-ol (0.5%), linalool oxide (0.09%), phenylethyl alcohol (1.46%) and benzaldehyde (0.68%).

## Sex and genetics influence the effects of Arabian jasmine on exploration of a novel tank

To determine if sex and genetics influence the behavioral effects of Arabian jasmine, we did a dose response (0, 5, 10, 20 mg kg$^{-1}$) in two strains of fish (AB and WIK) and both sexes in novel tank test (NTT) and mirror biting test (MBT). Behavior in the NTT was used to identify changes in anxiety-like behaviors (e.g., bottom distance) and locomotor activity, and the MBT was used to identify changes in aggressive behaviors. We assessed significance using 4 × 2 (dose × sex) ANOVAs followed by Dunnett's multiple comparison tests with the vehicle as the control group.

For the NTT, we assessed the effects of Arabian jasmine on bottom distance, center distance, percentage of tank explored, and distance travelled (Fig 2A). For bottom distance in AB zebrafish, we found a trend towards a small effect of dose ($p = 0.085$, $\eta^2 = 0.06$), a trend towards a small interaction ($p = 0.072$, $\eta^2 = 0.06$), and no effect of sex ($p = 0.50$). Dunnett's multiple comparison tests indicated large effects in the 10 mg kg$^{-1}$ ($p = 0.010$, $d = 1.22$, 95% CI [0.44, 1.98]) and 20 mg kg$^{-1}$ ($p = 0.038$, $d = 1.18$, 95% CI [0.40, 1.94]) female treatment groups where Arabian jasmine treated fish swam closer to the bottom of the tank than the vehicle group. For the bottom distance in WIK zebrafish (Fig 2A), we found no effect of dose ($p = 0.11$), sex ($p = 0.80$), or an interaction ($p = 0.59$). For the center distance in AB zebrafish (Fig 2A), we found a trend towards a small effect of sex ($p = 0.098$, $\eta^2 = 0.02$), but no effect of dose ($p = 0.88$), and no interaction

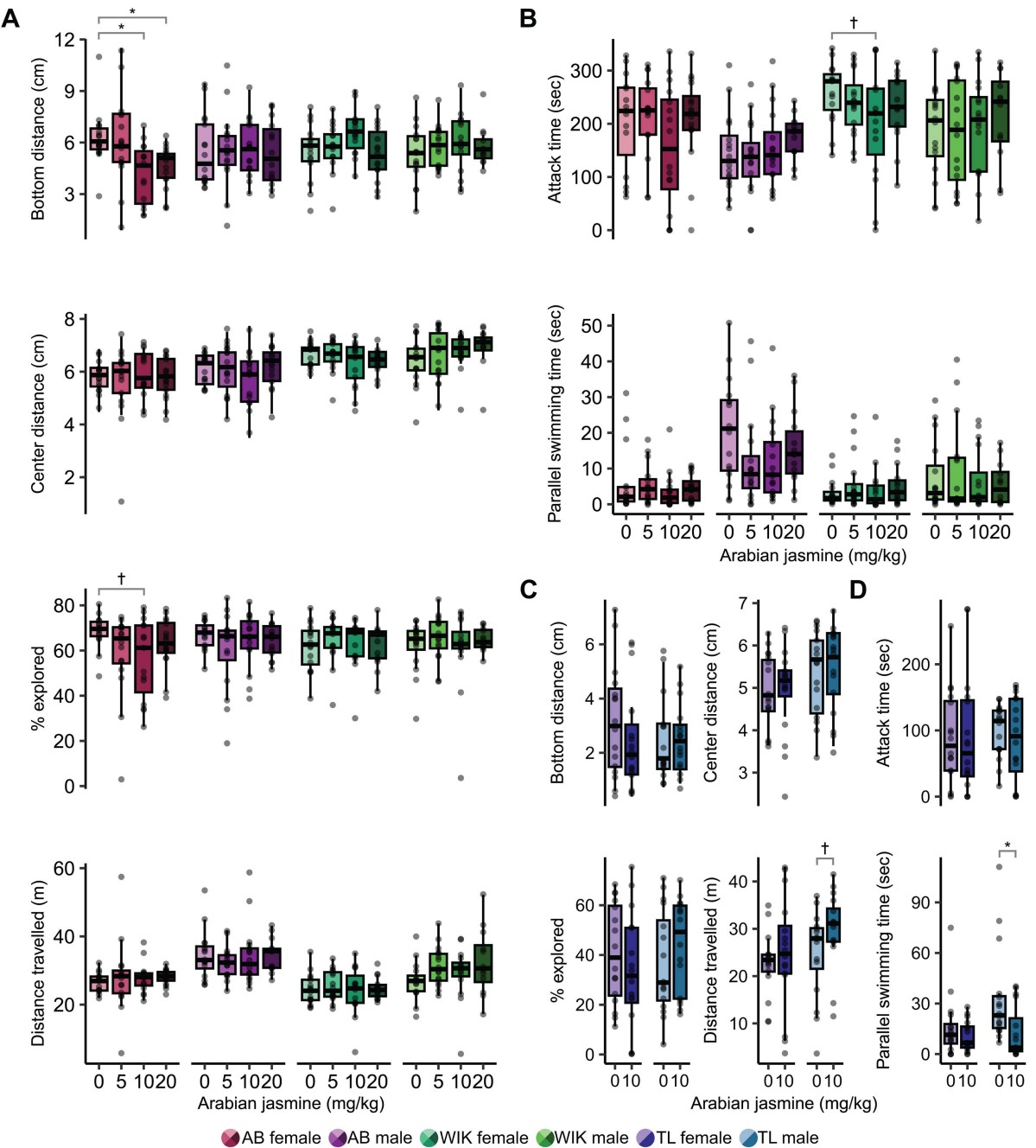

**Fig 2. Influence of sex and strain on the behavioral effects of Arabian jasmine.** The effect of Arabian jasmine on **(A)** exploratory behaviors (bottom distance, center distance, percent of exploration, and distance travelled) and **(B)** social behaviors (attack time and parallel swim time) in AB and WIK zebrafish based on dose (0, 5, 10,20 mg kg⁻¹) and sex. The effect of Arabian jasmine on **(C)** exploratory behaviors and **(D)** social behaviors in TL zebrafish given either vehicle or 10 mg kg⁻¹ Arabian jasmine. Boxplots indicate median (center line), interquartile range (box ends), and hinge±1.5 times the interquartile range (whiskers). *$p < 0.05$, †$p < 0.10$ from Dunnett's post-hoc in comparison to the 0 mg kg⁻¹ group (Figs 2A and 2B). *$p < 0.05$, †$p < 0.10$ from pairwise $t$-test with FDR correction within group (Figs 2C and 2D). AB females: $n$'s = 15,16,16,16 for 0, 5,10, and 20 mg kg⁻¹, respectively. AB males: $n$'s = 16,16,16, 14 for 0,5,10, and 20 mg kg⁻¹, respectively. WIK female: $n$'s = 16,16,15,15 for 0,5,10, and 20 mg kg⁻¹, respectively. WIK male: $n$'s = 16,16,16,12 for 0,5,10, and 20 mg kg⁻¹, respectively. TL female and males: $n$'s = 16.

($p = 0.47$). Dunnett's multiple comparison test did not show a significant difference from the control group. For the center distance in WIK zebrafish (Fig 2A), we found no effect of dose ($p = 0.87$), sex ($p = 0.22$), or an interaction ($p = 0.16$). For the percentage of exploration in AB zebrafish (Fig 2A), we found a trend towards a small effect of dose ($p = 0.064$, $\eta^2 = 0.06$), but no effect of sex ($p = 0.29$), or an interaction ($p = 0.51$). Dunnett's multiple comparison test indicated that female fish eating 10 mg kg$^{-1}$ had a trend towards a large effect of less exploration compared to the vehicle group ($p = 0.063$, $d = 0.90$, 95% CI [0.15, 1.63]). For the percentage of exploration in WIK zebrafish (Fig 2A), we found no effect of dose ($p = 0.60$), sex ($p = 0.92$), or an interaction ($p = 0.86$). Finally, for the distance travelled, in AB zebrafish (Fig 1A), we found a large effect of sex ($p < 0.001$, $\eta^2 = 0.20$), where males swam further than females, but no effect of dose ($p = 0.91$), or an interaction ($p = 0.74$). In WIK zebrafish (Fig 2A), we also found a large effect of sex ($p < 0.001$, $\eta^2 = 0.16$), with males travelling further than females, but no effect of dose ($p = 0.28$), or an interaction ($p = 0.42$).

During the MBT, we assessed both attacking and parallel swimming time (Fig 2B). For the attacking time in AB zebrafish (Fig 2B), we found a medium-sized effect of sex ($p = 0.004$, $\eta^2 = 0.07$), where females attacked more than males, but no effect of dose ($p = 0.36$), or an interaction ($p = 0.22$). For attack time in WIK zebrafish (Fig 2B), we again found a small effect of sex ($p = 0.014$, $\eta^2 = 0.05$), but no effect of dose ($p = 0.59$), or an interaction ($p = 0.37$). Finally, for the parallel swimming time in AB zebrafish (Fig 2B), we found a large effect of sex ($p < 0.001$, $\eta^2 = 0.22$), where males engaged in more parallel swimming than females, and a trend towards a small effect of dose ($p = 0.094$, $\eta^2 = 0.05$), but no interaction ($p = 0.44$). However, Dunnett's multiple comparison test did not find any significant differences from the vehicle control in either males or females. For parallel swimming time in WIK zebrafish (Fig 2B), we found a trend towards a small effect of sex ($p = 0.066$, $\eta^2 = 0.03$), but no effect of dose ($p = 0.63$), or an interaction ($p = 0.83$).

From the results of the dose-response experiment in NTT and MBT, we found that the 10 mg kg$^{-1}$ treatment group had an anxiogenic effect in female AB zebrafish but no effect in WIK zebrafish suggesting that the effects of this dose are influenced by genetic variation. Thus, we examined the effect of 10 mg kg$^{-1}$ Arabian jasmine on zebrafish of the TL strain as this strain has been found to have higher baseline levels of anxiety than other strains [27]. For bottom distance in TL zebrafish (Fig 2C), we found no effect of Arabian jasmine ($p = 0.49$), sex ($p = 0.43$), and no interaction ($p = 0.38$). For the center distance of TL zebrafish, we found a trend towards a small effect of sex ($p = 0.089$, $\eta^2 = 0.05$), but no effect of Arabian jasmine ($p = 0.79$), or an interaction ($p = 0.88$). For the percentage of exploration in TL zebrafish, we found no effect of Arabian jasmine ($p = 0.97$), sex ($p = 0.52$), and there was no interaction ($p = 0.17$). For the distance travelled of TL zebrafish, we found no effect of Arabian jasmine ($p = 0.12$), sex ($p = 0.12$), and no interaction ($p = 0.52$). However, post-hoc tests found that the 10 mg kg$^{-1}$ male treatment group ($p = 0.10$, $d = 0.59$, 95% CI [−0.12, 1.29]) had a trend towards a medium effect of higher activity than the vehicle group.

For the attacking time of TL zebrafish (Fig 2D), we found no effect of Arabian jasmine ($p = 0.77$), sex ($p = 0.89$), and no interaction ($p = 0.68$). For the parallel swimming time of TL zebrafish, we found a medium effect of Arabian jasmine ($p = 0.009$, $\eta^2 = 0.11$), a trend towards a medium effect of sex ($p = 0.051$, $\eta^2 = 0.06$), but no interaction ($p = 0.12$). Post-hoc tests found that the 10 mg kg$^{-1}$ male treatment group ($p = 0.017$, $d = 0.89$, 95% CI [0.15, 1.61]) had less parallel swimming time than the vehicle treated group.

### The effect of Arabian jasmine on anxiety-related behaviors is influenced by zebrafish personality

Our first experiments found that TL and WIK zebrafish had minimal responses to Arabian jasmine in the NTT, whereas AB zebrafish had an anxiogenic response at 10 mg kg$^{-1}$. We noticed a wide variation in behavioral response to Arabian jasmine in the AB fish and hypothesized that perhaps the personality of the animals may affect how they respond to Arabian jasmine. To test this, we examined two different personality-related behaviors: boldness as measured by a combination of bottom distance and percent of tank explored [15,27], and activity via distance travelled [28]. To determine personality type, we performed the NTT in the absence of Arabian jasmine on day 1, then, on day 2, individuals were given 10 mg kg$^{-1}$ Arabian jasmine to determine their response in the NTT and MBT. We assessed significance using 2 × 2

(treatment × boldness/activity level) ANOVAs within each sex followed by FDR-corrected pair-wise *t*-tests to compare groups. One-sample *t*-tests were used to determine if the behavior changed from day 1 to day 2. The distribution of the boldness index in each sex is presented in Fig 3A, and the distribution of the activity index in each sex is presented in Fig 3B. We found no correlation between activity and boldness, suggesting these measures are capturing distinct aspects of zebrafish behavior (Fig 3C).

First, we determined if boldness influences the behavioral response of zebrafish to Arabian jasmine (Figs 4A & S1A). For the bottom distance of AB female zebrafish, we found a medium effect of Arabian jasmine ($p = 0.045$, $\eta^2 = 0.06$), and boldness ($p = 0.030$, $\eta^2 = 0.07$), but no interaction ($p = 0.60$). Post-hoc tests found that the Arabian jasmine had a trend towards a small anxiolytic effect where they increased their bottom distance ($p = 0.056$, $d = 0.65$, 95% CI [−0.02, 1.31]). For AB male zebrafish, we found a large effect of boldness on bottom distance ($p = 0.0011$, $\eta^2 = 0.14$), but no effect of Arabian jasmine ($p = 0.86$), or an interaction ($p = 0.47$). For the change in bottom distance over time (Fig 4B), a one-sample *t*-test found that only shy female zebrafish, irrespective of drug treatment, increased distance from bottom ($p = 0.005$, $d = 0.69$, 95% CI [0.12, 1.26] for vehicle treated, and $p < 0.001$, $d = 1.24$, 95% CI [0.62, 1.86] for Arabian jasmine treated).

For center distance (Fig 4A), in female AB zebrafish, we found no effect of Arabian jasmine ($p = 0.36$), boldness ($p = 0.80$), and no interaction ($p = 0.56$). In male AB zebrafish, we found a medium effect of Arabian jasmine ($p = 0.017$, $\eta^2 = 0.08$), but no effect of boldness ($p = 0.61$), or an interaction ($p = 0.98$). Post-hoc tests found a trend toward medium size effects in Arabian jasmine treated bold males ($p = 0.070$, $d = 0.63$, 95% CI [−0.05, 1.30]) and shy males ($p = 0.10$, $d = 0.56$, 95% CI [−0.11, 1.23]), both swimming closer to the center of the tank. For the change in center distance over time (Fig 4B), a one-sample t-test found that the Arabian jasmine shy female treatment group swam closer to the border of the tank on the second day ($p = 0.049$, $d = 0.30$, 95% CI [0.00, 0.45]).

For the percentage of exploration of female AB zebrafish (Fig 4A), we found a medium sized effect of Arabian jasmine ($p = 0.030$, $\eta^2 = 0.07$), and boldness ($p = 0.007$, $\eta^2 = 0.10$), but no interaction ($p = 0.40$). Post-hoc tests found that Arabian jasmine caused shy female fish to explore the tank more than the vehicle group ($p = 0.016$, $d = 0.84$, 95% CI [0.16, 1.51]). For AB male zebrafish, we found a medium effect of boldness ($p = 0.007$, $\eta^2 = 0.10$), but no effect of Arabian jasmine ($p = 0.38$), or an interaction ($p = 0.59$). For the change in percentage of exploration over time (Fig 4B), the Arabian jasmine treated shy female group explored the tank more than day 1 ($p = 0.029$, $d = 0.53$, 95% CI [0.74, 12.24]).

For distance travelled (Fig 4A), in AB female zebrafish we found no effect of Arabian jasmine ($p = 0.66$), boldness ($p = 0.56$), and no interaction ($p = 0.17$). For distance travelled in males, we also found no effect of Arabian jasmine ($p = 0.31$), boldness ($p = 0.91$) and no interaction ($p = 0.20$). For the change in distance travelled over time (Fig 4B), we found that Arabian jasmine in bold females had a trend towards more activity than day 1 ($p = 0.062$, $d = 0.45$, 95% CI [−0.04, 0.94]).

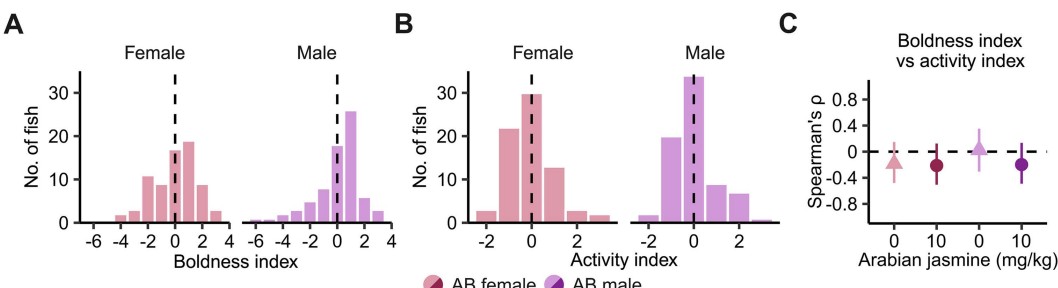

**Fig 3. Personality profiling in AB zebrafish based on boldness and activity. (A)** Histogram of boldness index, and **(B)** activity index on day 1 of exposure to the novel tank. The dashed line is the median. **(C)** Spearman's rank correlation coefficient (ρ) with 95% confidence intervals across boldness index and activity index. Female: $n = 72$. Male: $n = 73$.

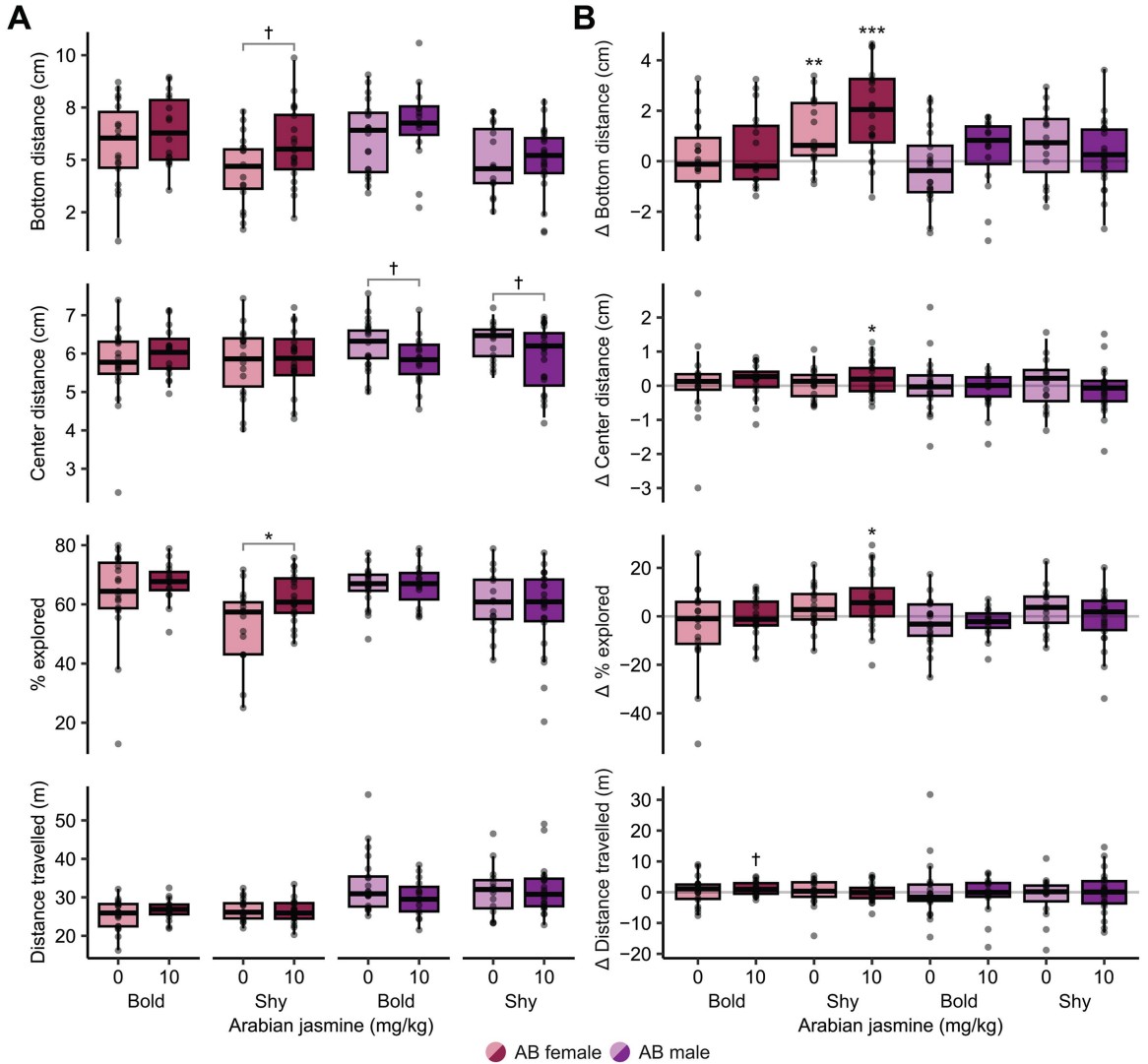

**Fig 4. Influence of boldness on the behavioral effects of Arabian jasmine. (A)** Exploratory behaviors on day 2 in bold animals given either vehicle or 10 mg kg⁻¹ Arabian jasmine. **(B)** Change in behavior between the first and second days of exploratory behaviors in bold animals given either vehicle or 10 mg kg⁻¹ Arabian jasmine. Boxplots indicate median (center line), interquartile range (box ends), and hinge±1.5 times the interquartile range (whiskers). *$p < 0.05$, †$p < 0.10$ from pairwise t-test with FDR correction within group. ***$p < 0.001$, **$p < 0.01$, *$p < 0.05$, †$p < 0.10$ compared to zero using one-sample t-tests with FDR corrections. Bold female: $n = 19$ vehicle (0 mg kg⁻¹ Arabian jasmine); $n = 16$ Arabian jasmine (10 mg kg⁻¹ Arabian jasmine). Shy female: $n = 17$ vehicle; $n = 20$ Arabian jasmine. Bold male: $n = 21$ vehicle; $n = 15$ Arabian jasmine. Shy male: $n = 15$ vehicle; $n = 22$ Arabian jasmine.

Next, we determined if activity levels influence the behavioral effects of Arabian jasmine in the NTT and MBT. There were no significant effects in the MBT (S1B Fig). In the NTT (Fig 5A), for bottom distance in female zebrafish, we found a medium sized effect of Arabian jasmine ($p = 0.046$, $\eta^2 = 0.06$), but no effect of activity ($p = 0.10$), and no interaction ($p = 0.25$). Post-hoc tests found that the low-activity female treatment group had a trend towards a medium-sized effect of bottom distance being higher than the vehicle group ($p = 0.054$, $d = 0.66$, 95% CI [−0.01, 1.32]). For males, we found a medium-sized effect of activity ($p = 0.025$, $\eta^2 = 0.07$), a trend towards a small interaction ($p = 0.076$, $\eta^2 = 0.05$), but no effect of treatment ($p = 0.86$). However, post-hoc tests found no differences between groups. For the change in bottom distance over time (Fig 5B), we found an effect of Arabian jasmine in females in both the high ($p = 0.002$, $d = 1.01$, 95% CI [0.62,

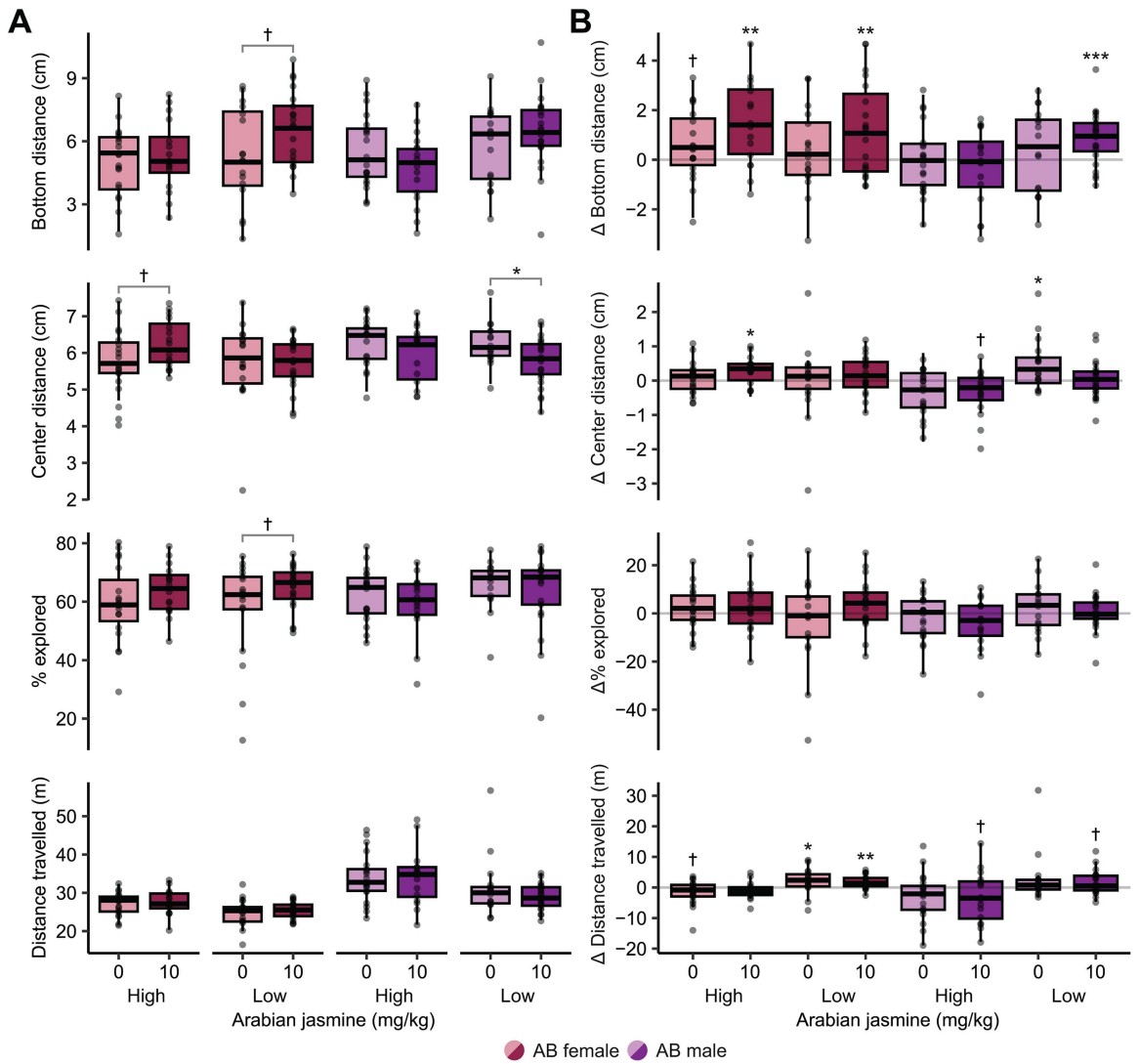

**Fig 5. Influence of activity on the behavioral effects of Arabian jasmine. (A)** Exploratory behaviors on day 2 based on activity levels in animals given either vehicle or 10 mg kg⁻¹ Arabian jasmine. **(B)** Change in behavior between the first and second days of exploratory behaviors based on activity levels in animals given either vehicle or 10 mg kg⁻¹ Arabian jasmine. Boxplots indicate median (center line), interquartile range (box ends), and hinge±1.5 times the interquartile range (whiskers). $*p < 0.05$, $†p < 0.10$ from pairwise t-test with FDR correction within group. $***p < 0.001$, $**p < 0.01$, $*p < 0.05$, $†p < 0.10$ compared to zero using one-sample t-tests with FDR corrections. High activity females: $n = 19$ vehicle (0 mg kg⁻¹ Arabian jasmine); $n = 16$ Arabian jasmine (10 mg kg⁻¹ Arabian jasmine). Low activity females: $n = 17$ vehicle; $n = 20$ Arabian jasmine. High activity males: $n = 20$ vehicle; $n = 16$ Arabian jasmine. Low-activity males: $n = 16$ vehicle; $n = 21$ Arabian jasmine.

2.39]) and low activity ($p = 0.010$, $d = 0.68$, 95% [0.25, 2.03]) groups. In the male low-activity group, Arabian jasmine also had an effect ($p < 0.001$, $d = 0.50$, 95% CI [0.37, 1.36]), causing a shift to higher distance from the bottom on the second day. The vehicle treated high-activity group had a trend towards changing to a higher distance from the bottom ($p = 0.070$, $d = 0.44$, 95% CI [−0.06, 1.33]).

For the center distance of AB female zebrafish (Fig 5A), we found no effect of Arabian jasmine ($p = 0.35$), activity ($p = 0.21$), and no interaction ($p = 0.21$). For the center distance of AB male zebrafish, we found a medium effect of Arabian jasmine ($p = 0.016$, $\eta^2 = 0.08$), but no effect of activity ($p = 0.34$), or an interaction ($p = 0.61$). Post-hoc tests found that

Arabian jasmine treated low-activity males swam closer to the center of the tank compared to the vehicle group ($p=0.046$, $d=0.69$, 95% CI [0.02, 1.36]). For the change in center distance over time (Fig 5B), there was an effect in Arabian jasmine treated high-activity females ($p=0.02$, $d=0.46$, 95% CI [0.06, 0.48]) and vehicle treated low-activity males ($p=0.02$, $d=0.86$, 95% CI [0.09, 0.84]) where they were both closer to the border of the tank. There was also a trend towards an effect in Arabian jasmine treated high-activity males ($p=0.06$, $d=0.53$, 95% CI [−0.73, 0.02]).

For the percentage of exploration in female AB zebrafish (Fig 5A), we found a medium-sized effect of Arabian jasmine ($p=0.040$, $\eta^2=0.06$), but no effect of activity ($p=0.83$), or an interaction ($p=0.61$). Post-hoc tests found that the Arabian jasmine treated low-activity females had a trend toward a medium effect of exploring the tank more than the vehicle group ($p=0.091$, $d=0.57$, 95% CI [−0.09, 1.23]). For the percentage of exploration of male AB zebrafish, we found no effect of Arabian jasmine ($p=0.40$), activity ($p=0.12$), and no interaction ($p=0.86$). For the change in percentage of exploration over time (Fig 5B), we found no change in exploration behavior in either vehicle or Arabian jasmine groups.

For the distance travelled of female AB zebrafish (Fig 5A), we found a medium-sized effect of activity ($p=0.004$, $\eta^2=0.12$), but no effect of Arabian jasmine ($p=0.64$), or interaction ($p=0.91$). For the distance travelled of male AB zebrafish, we found a medium-sized effect of activity ($p=0.011$, $\eta^2=0.09$), but no effect of Arabian jasmine ($p=0.29$), or an interaction ($p=0.37$). Post-hoc tests found no differences for either females or males. For the change in distance travelled over time (Fig 5B), we found an effect in vehicle ($p=0.044$, $d=0.72$, 95% CI [0.06, 4.27]) and Arabian jasmine ($p=0.003$, $d=0.89$, 95% CI [0.68, 2.74]) treated low-activity females, both of which had more activity than day 1. The vehicle treated high-activity females ($p=0.090$, $d=0.56$, 95% CI [−3.63, 0.28]) and Arabian jasmine treated low-activity males ($p=0.080$, $d=0.51$, 95% CI [−0.21, 3.35]) had trends toward less activity than day 1. Arabian jasmine treated high-activity males had a trend toward less activity ($p=0.090$, $d=0.51$, 95% CI [−6.54, 0.53]).

## Discussion

We find that sex, genetic background, and personality all contribute to the anxiety-related behavioral effects of Arabian jasmine in adult zebrafish. This conclusion is based on the following: (1) Arabian jasmine caused a decrease in bottom distance, and to a lesser extent percent explored, (i.e., was anxiogenic) in female AB fish (Fig 2A); there was no effect in AB males, and no effect in WIK or TL fish of either sex. (2) Arabian jasmine increased percent explored, and to a lesser extent bottom distance (i.e., was anxiolytic) in shy, but not bold, female AB fish (Fig 4A), and decreased center distance in male AB fish with low activity (Fig 5). Taken together, these data suggest that strain, sex, and personality interact to modulate the effects of Arabian jasmine on behavior.

Our paper is the first to find that Arabian jasmine affects behavior in zebrafish. Given that Arabian jasmine is a complex mixture of many phytochemicals, it is not clear which constituents may be driving these behavioral effects. Chemical analysis of our preparation may provide some clues: We had particularly high concentrations of linalool (28.4%) and dimethyl sulfide (30.3%). We also isolated lesser amounts of other potentially psychoactive compounds such as 3-hexen-1-ol (0.5%) [29], linalool oxide (0.09%) [30], phenylethyl alcohol (1.46%) [31] and benzaldehyde (0.68%) [11]. Most of these compounds have been described as anxiolytic [8–10,29–34] except for benzaldehyde [11]. However, there are no reports on the behavioral effects of dimethyl sulfide, which is known for having an unpleasant odor [35].

The high concentration of linalool in our extract is notable because it has been consistently found to be anxiolytic [10,32–34,36]. This may be via the many effects linalool has been found to have on neurotransmission. For example, linalool enhances GABAergic currents in mice [36] and directly inhibits glutamatergic neurons [37,38]. Linalool has also been found to modulate serotonergic functioning, where Jarvis et al. [33] found that linalool acts as a 5-HT$_3$ antagonist. 5-HT$_3$ receptors are ionotropic receptors that mediate excitatory neurotransmission; their activity has been linked to anxiety [39,40]. Linalool affects other neurotransmitter systems as well, like dopamine and norepinephrine [32]. At the physiological level, Höferl et al. [34] found linalool to promote relaxation in humans alongside a reduction in cortisol and heart rate, and Yamamoto et al. [10] found restrained mice that inhaled linalool had reduced hypothalamic-pituitary-adrenal

(HPA) activity. Behaviorally, linalool was found to be anxiolytic in mice [32,41] and to cause a reduction in aggression in male mice [9]. Taken together with the present study, these findings suggest that the anxiolytic effects of Arabian jasmine may be mediated by linalool's effects on neurotransmitters and stress hormone activity in zebrafish.

In contrast to linalool, benzaldehyde is one of the few compounds in Arabian jasmine that has been described as anxiogenic, although the evidence here is weaker as there is only one study describing its effects. Laham et al. [11] found that inhalation of benzaldehyde by Sprague-Dawley rats resulted in increased aggressive behavior, noise sensitivity, and reduced motor activity. Interestingly, Tabatabaie and Floyd [42] found that benzaldehyde inactivates the antioxidant enzyme glutathione peroxidase, which can disrupt the redox balance; this may lead to elevated anxiety via an increase in oxidative stress [43]. Thus, it may be the case that the anxiogenic effects we observed in female AB zebrafish are due to higher sensitivity of these fish to benzaldehyde. However, given the complex mixture of chemicals in Arabian jasmine, more research would be needed to more strongly draw this conclusion.

There are some notable differences in the chemicals we identified in Arabian jasmine compared to previous reports. For example, we find higher linalool (28.4%) than prior work (2–10%) and dimethyl sulfide has not previously been reported to be a constituent of *J. sambac* [6,7,44]. These differences may be because we used ultra-sonic assisted extraction where microbubbles break down plant walls [45]. This contrasts with other methods that use warm water [1] that can degrade volatile compounds [46]. Additionally, our air pressure-assisted extraction may enhance this process by improving mass transfer efficiency [47]. Thus, compared with prior methods, our approach minimizes the loss of volatile compounds. Different preparation methods are a challenge for the field of traditional medicine as it has been found that the intensity, solvent choice, and temperature used in ultrasonic-assisted extraction can cause the breakdown of active compounds [45]. For example, the high concentration of dimethyl sulfide we observed may have been produced through a Maillard reaction between glucose and methionine in the plant, with ultrasonic treatment accelerating the degradation of methionine and promoting the formation of this volatile sulfur compound [48,49]. More work is clearly needed to better understand the impact of extraction techniques on the phytochemical profile and subsequent behavioral effects of Arabian jasmine.

We chose a gelatin-based feed to administer Arabian jasmine to zebrafish [50] because it is less stressful than alternatives like beaker dosing and intraperitoneal injections, and it allows us to dose fish based on their weight. By dosing fish based on weight, we can directly compare between fish and other common model organisms, like rats and mice, that take a similar approach. Although there is a potential for the leaching of jasmine into the water, we have found that leaching is minimal if animals eat the feed within 1 minute; often they eat the feed within 5–10 seconds once acclimated [50]. Finally, people typically consume Arabian jasmine as an infused drink [51,52], and thus our gelatin feed more closely matches this route of administration.

We found several instances in which genetic background and sex influence the effects of Arabian jasmine on behavior. Arabian jasmine caused female AB zebrafish to decrease their bottom distance and exploratory behavior, while there was no effect in males, and no effect in WIK or TL fish of either sex. Genetic variation is known to influence a wide variety of behavioral responses [26,27,53,54]. This is likely due, in part, to differences in neurochemical profiles [55–57] that may affect drug responses. For example, strain specific effects of anxiolytics, like diazepam and losartan, have been observed in mice where diazepam induced anxiolytic effects in C57BL/6, DBA/2, and BKW mice whereas losartan produced effects only in BKW mice [58]. Similarly, Dlugos and Robin [59] found that strain affected behavioral responses to ethanol in WT, LFS and BLF zebrafish where only they observed anxiolytic effects in the WT and LFS strains. Beyond strain differences, sex also plays a significant role in drug responses. Male and female zebrafish differ in their responses to ethanol, where females are more sensitive than males [12,60]. Johnson et al. [61] also investigated the effect of chondroitin sulfate on swimming velocity in zebrafish, finding increased velocity only in males. There are also several baseline differences in behavior between the sexes. For example, Fontana et al. [62] identified sex differences in anxiety-related behaviors in male and female zebrafish, and Rajput et al. [27] used a three-dimensional novel tank test to observe sex-based differences in anxiety-related behaviors, including distance from the center, distance traveled, and percentage of exploration.

Sex differences are also seen at the neurochemical level in zebrafish where Beigloo et al. [15] found that males have higher dopamine and serotonin than females. Thus, the differential responses to Arabian jasmine observed in this study is likely driven by an interaction of genetics and sex, possibly due to variations in in neurotransmitter systems.

We found that different behavioral types (i.e., personality) in zebrafish also influenced the anxiety-related response of Arabian jasmine. For example, we found Arabian jasmine increased exploratory behaviors in shy females and elevated bottom distance in low-activity males. Variations in drug responses based on personality types have been observed in mice and zebrafish in a handful of cases. For instance, Beigloo et al. [15] found differences in the responses of bold and shy WIK zebrafish following escitalopram administration. In mice, Mehrhoff et al. [16] found that diazepam had an anxiolytic effect only low-activity mice. One possible explanation for these effects is differences in gene expression that may underlie behavioral types. For example, Norton et al. [63] identified *fgfr1a* as a key gene influencing boldness behavior in zebrafish. Similarly, Thomas et al. [64] studied genetic variation in inbred mice with high and low open-field activity and identified *Ctsc* and *Vmn1r1* as genes linked to differing activity levels. Finally, Booher et al. [65] showed that *Ndufa13* gene encoding the supernumerary subunit A13 of nicotinamide adenine dinucleotide dehydrogenase complex I, is a key candidate gene influencing high or low activity in mice. Thus, the varied responses to Arabian jasmine based on personality-type in zebrafish we observed may be driven by associated interindividual differences in gene expression.

In this study, we observed conflicting results: the 10 mg kg$^{-1}$ Arabian jasmine treatment group in female AB zebrafish exhibited more anxiety-like behavior, while Arabian jasmine had an anxiolytic effect in shy female AB zebrafish. This phenomenon, known as a paradoxical drug effect, has been reported with anxiolytic drugs such as benzodiazepines [66] and fluoxetine [67]. For example, in patients who have genetic links to psychological disorders, anxiogenic effects of benzodiazepines were more likely [66]. In addition, the age of mice can influence responses to fluoxetine, with anxiolytic effects in adults and anxiogenic effects in juvenlies [67]. One possible explanation for this paradoxical effect could be differences in HPA axis function. Lindholm et al. [68] found that genetic variation can influence the functioning of the HPA axis and associated neurochemicals, potentially contributing to paradoxical drug responses. Beigloo et al. [15] also found the differences in serotonergic system can affect the boldness of female zebrafish, which could help explain these unexpected drug responses. In mice, Krakenberg et al. [69] showed that variations in the 5-HT transporter gene were associated with different levels of anxiety-related behavior in mice. The level of GABA may also play an important role in paradoxical drug effects. Majewska [70] found that variation in neuro-steroid levels can influence GABA$_A$ receptor activity. These findings suggest that differing neurochemical profiles in fish of different background strains or different personality types may contribute to the paradoxical response to Arabian jasmine in zebrafish.

This study has some limitations. For example, we administered Arabian jasmine acutely, with a single dose, whereas humans typically consume it chronically [1]. Additionally, humans often use herbal preparations containing a combination of multiple plants. This study selected a dose of Arabian jasmine based on its use in such preparations; however, for simplicity of interpretation, the effects of interactions with other plants were not examined. We also did not examine biochemical correlates of anxiety, like HPA axis function. Future work should explore cortisol level to identify potential differences in HPA axis functioning across strain, sex, and personality that could contribute to the variable effects of Arabian jasmine [71]. Another factor that we did not examine here that could influence the effects of Arabian jasmine are differences in pharmacokinetic profiles across strains and sexes. Finally, it is difficult to extrapolate the anxiety-like behavior observed here to the phenomenological experience of humans taking Arabian jasmine. Nonetheless, we believe our work lays the foundation for using zebrafish to unravel the biological basis for the complex behavioral effects of Arabian jasmine and other natural compounds used in traditional medicine.

## Conclusion

In summary, our work reveals the effects of Arabian jasmine on anxiety-related behavior is affected by background genetics, sex and personality. Further work is needed to explore the mechanisms of action at the neural and molecular levels.

Nonetheless, our results lead to the conclusion that the paradoxical responses of Arabian jasmine that have previously been reported may be due to differences in the characteristics of subjects.

## Methods

### Plant materials

*Jasminum sambac* (L.) Aiton (Fig 1A) was collected from Arabian jasmine cultivation in Khumphaeng Phet, Thailand (16°27'45.0"N 99°40'11.3"E) on the morning (8–10AM) of 25th April 2023. Voucher specimens (herbarium number: PBM006430) were made using standard herbarium specimen preparation procedures [72] and deposited in the herbarium at Sireeruckhachati Nature Learning Park, Mahidol University, Thailand. The Flora of China [73] and the Flora of Thailand [74], as well as a recent document of the genus *Jasminum* in Thailand [75], were used for identification.

### Plant extraction and chemical identification

To replicate the preparation used in traditional medicine, water was chosen as the solvent. This process used ultrasonic- and pressure-assisted extraction methods (Fig 1B). The ultrasonic-assisted extraction method was adapted from Vinatoru et al. [76], while the pressure-assisted extraction was modified from Ong et al. [77]. Arabian jasmine extract was prepared by sonicating Arabian jasmine flowers in water at a ratio of 5 parts Arabian jasmine to 3 parts water by weight for 15 minutes. The resulting aqueous solution was then subjected to continuous extraction under reduced air pressure (400 mmHg) using a rotary evaporator for 30 minutes. After filtration, the extract was stored in an airtight, light-protected container at −20°C. To create a concentrated solution, 100 mL of the Arabian jasmine extract was evaporated using a rotary evaporator to yield a concentrate of 25 mg mL$^{-1}$.

Following the extraction, a gas chromatograph equipped with a headspace extractor (7697A Static Headspace Sampler and Agilent 7890B System, Agilent Technologies, Santa Clara, CA, USA) was used for chemical analysis. The system included an HP-INNOWax capillary column (30 m length × 250 µm diameter × 0.25 µm thickness, Agilent Technologies) and a mass spectrometer with an ion trap detector (Agilent Technologies). The analysis was performed in the range of 33–400 m/z under ionization energy of 70 eV. n-Alkane compounds were used as external reference standards to ensure accuracy.

The chromatographic analysis of the Arabian jasmine extract (Fig 1C) was conducted by comparing retention times, peak areas, peak heights, and mass spectral patterns with those in the NIST library [78] of authentic compounds.

### Gelatin preparation and gelatin-feed habituation

For gelatin feed preparation, we followed the method from Ochocki and Kenney [50]. Arabian jasmine water extract was administered using a gelatin-based feed at dose of 5, 10, or 20 mg kg$^{-1}$ body mass. This was determined to approximately match the suggested doses in traditional preparations given to people [1]. The gelatin feed consisted of 12% w/v gelatin (Sigma-Aldrich), 4% w/v spirulina (Argent Aquaculture, Redmond, WA, USA) and brine shrimp extract. The extract was made by suspending 250 mg ml$^{-1}$ of micro fine brine shrimp (Brine Shrimp Direct) in water followed by 1 h of stirring. The suspension was then centrifuged twice at 12,500 *g*, keeping the supernatant each time, and then diluted in two volumes of water before addition to the gelatin feed mixture. Arabian jasmine and water were added before warming the solution to 45°C. After warming, drug- or vehicle-containing solution was pipetted into individually sized morsels for feeding at 1% body mass. Gelatin was allowed to set on ice for at least 20 min before feeding.

Fish were individually habituated to the gelatin-feed for three days before behavior tests. On each day of the experiment, individuals were removed from the housing racks to the behavioral room for 1 h before being given a non-dosed gelatin feed instead of their morning feed. Individuals were isolated by the placement of transparent barriers in the tanks for 1 h. Once feed was given, we recorded whether individuals ate the feed within 5 min.

## Subjects

Subjects were female and male zebrafish of the AB, TL and WIK strains. We chose these strains due to their widespread usage and availability from high quality vendors. All fish were within two generations of breeders obtained from the Zebrafish International Resource Center (ZIRC; University of Oregon). Fish were 4–6 months of age and raised at Wayne State University (WSU). All procedures were approved by the Wayne State University Institutional Animal Care and Use Committee (Protocol ID: 21-02-3238). Housing was in high-density racks under standard conditions (water temperature: 27.5 ± 0.5°C, salinity: 500 ± 10 µS and pH 7.4 ± 0.2) with a 14 h:10 h light: dark cycle (lights on at 08:00 h). Fish were fed twice daily, in the morning with a dry feed (Gemma 300, Skretting, Westbrook, ME, USA) and in the afternoon with brine shrimp (*Artemia salina*, Brine Shrimp Direct, Ogden, UT, USA). The sex of fish was determined using three secondary sex characteristics: shape, color and presence of pectoral fin tubercles [79]. After experiments, animals were euthanized via immersion in ice cold water. Sex was confirmed following euthanasia by the presence or absence of eggs.

Fish were tested in three experiments. The initial dose–response experiment was conducted using AB and WIK strains (n ≈ 16 per group), though final sample sizes varied slightly due to gelatin-feeding refusal or pre-test mortality. Based on significant behavioral differences observed at 10 mg kg$^{-1}$ in the AB strain, this dose was selected for follow-up testing in the TL strain (n = 16 per group). As no behavioral differences emerged in TL, we then focused on individual differences in the AB strain, which had shown treatment sensitivity. In this final phase, zebrafish were assessed for personality traits (boldness and activity) using the same individuals. Overall, the selection of strains and group sizes across phases was guided by initial outcomes and the goal of identifying sources of inter-individual variation in response to treatment.

## Novel tank and mirror biting tests

We used the novel tank test to examine anxiety-related behavior, following the method described by Rajput et al. [27]. Additionally, the mirror biting test was employed to assess aggressive behavior.

One week before the experiment, fish were housed in 2 L tanks, with two female/male pairs per tank. Each tank was divided in half using a transparent divider, allowing one pair per side. To acclimate to the behavioral testing environment, tanks were removed from their housing racks and transferred to the behavioral room for 1 h. Three days before Arabian jasmine administration, fish were acclimated to the gelatin feed to ensure familiarity with the feeding method. Behavioral testing was conducted between 09:00 h and 14:00 h, with fish counterbalanced across sex and drug treatment groups. To avoid the buildup of chemical cues released by the fish during testing, NTT water was replaced between animals.

For the NTT, experimental tanks were five-sided (15 × 15 × 15 cm) and made from frosted acrylic (TAP Plastics, Stockton, CA, USA). Each tank was filled to a height of 12 cm with 2.5 L of fish facility water and placed in a white Plasticore enclosure to diffuse light and minimize external disturbances. Fish were individually placed into the tanks for 6 minutes while video recordings were made. Fish were tracked in the videos using DeepLabCut [80].

Following the NTT, fish were transferred to MBT tanks for another 6-minute session, during which video recordings were also made. The MBT tanks were identical to the NTT tanks, except for the inclusion of a mirror on the right-side wall. Like the NTT tanks, they were filled with 2.5 L of fish facility water and housed in a white Plasticore enclosure. D435 Intel RealSense™ depth-sensing cameras (Intel, Santa Clara, CA, USA) were mounted 20 cm above the tanks to record the sessions. These cameras were connected to Linux workstations via high-speed USB cables (NTC Distributing, Santa Clara, CA, USA). The MBT videos were first tracked using DeepLabCut [80] and then SimBA [81] was used to quantify social and aggressive behavior. We followed the behavioral terms of Blaser and Gerlai [82] (attack time as aggressive behavior) and Porfiri et al. [83] (parallel swimming time as social behavior). To measure attack time, we calculated the total time a fish spent attacking within 3 cm of the mirror (discrimination threshold: 0.35; minimum bout length: 20 ms). Similarly, to measure parallel swimming, we calculated the total time a fish spent swimming parallel to its reflection within 3 cm of the mirror (discrimination threshold: 0.4; minimum bout length: 50 ms).

## Dose response and genetic variation

After habituation to the gelatin-feed, individuals were given either vehicle or Arabian jasmine containing feed 30 min before being placed in the experimental tank. Only fish that ate the feed within 5 min were included in the analysis (3 female and 7 male fish given vehicle were excluded). We centered our dose response around the 10 mg kg$^{-1}$ dose based on the traditional preparation used for people [1]. Thirty minutes was chosen because several studies in zebrafish have found that oral drug administration results in peak serum concentrations within 40 min [84,85], and previous studies found behavioral response of drug administration using this method at this time point [50].

## Identifying personality type

For the effect of Arabian jasmine on the personality of zebrafish experiment, we followed the method of Beigloo et al. [15]. The behavior tests occurred over 2 days. On day 1, all individuals were given non-dosed feed 30 min before being placed in the novel tank. The boldness (z-scores of bottom distance and the percentage of exploration added together) and activity (z-score of distance travelled) categories were calculated from day 1. On day 2, individuals were given either vehicle or Arabian jasmine containing feed 30 min before being placed in the experimental tanks. The AB zebrafish and dose of 10 mg kg$^{-1}$ Arabian jasmine were selected because of the results of the dose-response experiment.

## Data analysis, visualization, availability

Data analysis was performed using R version 4.4.0 [86]. Graphs were made using ggplot2 [87]. Statistical analysis was done using a 4 × 2 (treatment × sex) ANOVA, a 2 × 2 (treatment × boldness/activity) ANOVA, or a one-sample *t*-test (the change of behavior over time). All tests were two-tailed. Omnibus tests were followed up with Dunnett's multiple comparison tests to examine the effects of Arabian jasmine dosage [88] and pairwise *t*-test with False Discovery Rate (FDR) correction to compare the effects of two different doses (0 and 10 mg kg$^{-1}$) within sex [89]. For effect sizes, ANOVAs are reported as $\eta^2$ and t-tests as Cohen's d. The interpretation of effect sizes as small ($0.01 < \eta^2 < 0.06$; $0.2 < d < 0.5$), medium ($0.06 \leq \eta^2 < 0.14$; $0.5 \leq d < 0.8$) or large ($\eta^2 \geq 0.14$; $d \geq 0.8$) based on Cohen [90].

## Supporting information

**S1 Fig. Influence of Arabian jasmine on aggressive behaviors based on personality.**
(DOCX)

**S1 Table. Composition of the Jasmine from GC-MS.**
(CSV)

## Acknowledgments

We thank Dinh Luong for the excellent maintenance and husbandry of our zebrafish colony.

## Author contributions

**Conceptualization:** Tripatchara Atiratana, Justin W. Kenney.

**Formal analysis:** Tripatchara Atiratana.

**Funding acquisition:** Tripatchara Atiratana, Nalena Praphairaksit, Justin W. Kenney.

**Investigation:** Tripatchara Atiratana.

**Methodology:** Tripatchara Atiratana, Aliyah R. Goldson, Siritron Samosorn, Neha Rajput, Justin W. Kenney.

**Project administration:** Justin W. Kenney.

**Supervision:** Siritron Samosorn, Nalena Praphairaksit, Justin W. Kenney.

**Visualization:** Tripatchara Atiratana.

**Writing – original draft:** Tripatchara Atiratana.

**Writing – review & editing:** Tripatchara Atiratana, Aliyah R. Goldson, Justin W. Kenney.

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
