## [Decision Letter · Decision Letter 0]

29 Jul 2025

PONE-D-25-26581The effects of Arabian jasmine on zebrafish behavior depends on strain, sex, and personalityPLOS ONE

Dear Dr. Kenney,

Thank you for submitting your manuscript to PLOS ONE. After careful consideration, we feel that it has merit but does not fully meet PLOS ONE’s publication criteria as it currently stands. Therefore, we invite you to submit a revised version of the manuscript that addresses the points raised during the review process.

**Both reviewers have judged the manuscript scientifically grounded and well written. Several valuable advises on how to further improve its quality are contained in their reports. Please make additional check in instruction for authors if the .eps would be acceptable figure format, or they should be rasterized.** ==============================

We look forward to receiving your revised manuscript.

Kind regards,

Branislav T. Šiler, Ph.D.

Academic Editor

PLOS ONE

Journal Requirements: 

3. To comply with PLOS ONE submissions requirements, in your Methods section, please provide additional information regarding the experiments involving animals and ensure you have included details on (1) methods of sacrifice, and (2) efforts to alleviate suffering.

 [National Institutes of General Medical Sciences (NIGMS; R35GM142566 to JWK)

Science Achievement Scholarship of Thailand (to TA and NP)]. 

[Funding was provided by NIGMS (R35GM142566 to JWK) and the Science Achievement Scholarship of Thailand (SAST to TA and NP). The funders had no role in study design, data collection and analysis, decision to publish, or preparation of the manuscript. We thank Dinh Luong for the excellent maintenance and husbandry of our zebrafish colony.]

  [National Institutes of General Medical Sciences (NIGMS; R35GM142566 to JWK)

Science Achievement Scholarship of Thailand (to TA and NP)]. 

6. Thank you for uploading your study's underlying data set. Unfortunately, the repository you have noted in your Data Availability statement does not qualify as an acceptable data repository according to PLOS's standards.

7. Please upload a copy of Supporting Information FigureTable S1 to S6 which you refer to in your text on page 21.

Reviewers' comments:

Reviewer's Responses to Questions

**Comments to the Author**

1. Is the manuscript technically sound, and do the data support the conclusions?

Reviewer #1: Yes

Reviewer #2: Yes

2. Has the statistical analysis been performed appropriately and rigorously? 

Reviewer #1: Yes

Reviewer #2: Yes

3. Have the authors made all data underlying the findings in their manuscript fully available?

Reviewer #1: Yes

Reviewer #2: Yes

4. Is the manuscript presented in an intelligible fashion and written in standard English?

Reviewer #1: Yes

Reviewer #2: Yes

5. Review Comments to the Author

Reviewer #1: I have reviewed manuscript entitled The effects of Arabian jasmine on zebrafish behavior depends on strain, sex, and personality. Manuscript provides novel data which are within the scope of the PLOS One journal. I would suggest minor correction which might contribute to the quality of the paper:

L73 and, and although – delete one and

L120 I would suggest that Figure 1 should be cited only in Methods section.

L125 Citations to literature in regards to different effects should be moved and/or kept only in discussion section.

Introduction should contain section framing the paradox of both antigenic and anxiolytic effects.

Clarify the ratio of selection of three used strains (AB, TL and WIK).

Please use confidence intervals to support interpretation of the near-significant findings in regards of the effects.

Reviewer #2: The work is well written, but I suggest further corrections below:

1) Pg 11, line 91. To correct: Zebrafish [Danio rerio, Hamilton (1922)];

2) Pg 11, lines 101 - 106. Remove this snippet: "Here, we find...". Al snippet is results/conclusion. Delete from introduction;

3) Rewrite the introduction with updated citations, i.e., from the last 5 years;

4) Pg. 12, lines 120-121: 120. Rewrite the snippet: "Arabian jasmine (Fig. 1A) was extracted via ultrasonic-assisted extraction with air pressure (Fig. 1B), using water as the solvent". The way it is written is method. Write in the form of result!

5) Put the "p" of statistical analysis in all text in lowercase;

6) Pg 21, lines 433-434. To correct: [e.g., serotonin transporter (5-HTT)

gene];

7) Increase the size of figures;

8) In the discussion explain the reason for preparing feed with gelatin;

6. PLOS authors have the option to publish the peer review history of their article (what does this mean? ). If published, this will include your full peer review and any attached files.

**Do you want your identity to be public for this peer review?** For information about this choice, including consent withdrawal, please see our Privacy Policy .

Reviewer #1: No

Reviewer #2: **Yes: ** Francisco Ernani Alves Magalhães

---

## [Author Response · Author response to Decision Letter 1]

22 Aug 2025

August 19, 2025

Dear Dr. Šiler,

Thank you for taking the time to have our manuscript reviewed and the opportunity to revise our manuscript entitled “The effects of Arabian jasmine on zebrafish behavior depends on strain, sex, and personality” (Manuscript ID: PONE-D-25-26581). We appreciate the thoughtful and constructive comments, which have helped us improve the clarity and quality of the manuscript.

In addition, this is our amended funding statement as requested:

Funding statement

National Institutes of General Medical Sciences (NIGMS; R35GM142566 to JWK

Science Achievement Scholarship of Thailand (to TA and NP).

Below, we provide detailed responses to each comment. All changes made in the manuscript have been tracked. Page and line numbers refer to the (clean) revised version.

Editorial comments

Comment 1: Please ensure that your manuscript meets PLOS ONE's style requirements, including those for file naming.

Response: Our manuscript is edited and met the PLOS ONE's style requirements followed these template (https://journals.plos.org/plosone/s/file?id=wjVg/PLOSOne_formatting_sample_main_body.pdf and https://journals.plos.org/plosone/s/file?id=ba62/PLOSOne_formatting_sample_title_authors_affiliations.pdf.).

The heading style is now used. Figure captions are now included in text when each figure is described for the first time. Supplementary data follow the guidelines.

Comment 2: In your Methods section, please provide additional information regarding the permits you obtained for the work. Please ensure you have included the full name of the authority that approved the field site access and, if no permits were required, a brief statement explaining why.

Response: Our manuscript contains information regarding the necessary approvals for animal research (lines 608-610). We note that this work was performed in a lab and not in the field, thus no permits were required. In the United States, where this work was performed, the only requirement is that the work is approved by an Institutional Animal Care and Use Committee (IACUC).

Comment 3: To comply with PLOS ONE submissions requirements, in your Methods section, please provide additional information regarding the experiments involving animals and ensure you have included details on (1) methods of sacrifice, and (2) efforts to alleviate suffering.

Response: We have now added a line indicating our method of euthanasia (lines 616-618), which was by immersion in ice cold water.

Comment 4: Thank you for stating the following financial disclosure:

[National Institutes of General Medical Sciences (NIGMS; R35GM142566 to JWK)

Science Achievement Scholarship of Thailand (to TA and NP)].

Response: We have included the amended funding statement above in our cover letter.

Comment 5: Thank you for stating the following in the Acknowledgments Section of your manuscript:

[Funding was provided by NIGMS (R35GM142566 to JWK) and the Science Achievement Scholarship of Thailand (SAST to TA and NP). The funders had no role in study design, data collection and analysis, decision to publish, or preparation of the manuscript. We thank Dinh Luong for the excellent maintenance and husbandry of our zebrafish colony.]

[National Institutes of General Medical Sciences (NIGMS; R35GM142566 to JWK)

Science Achievement Scholarship of Thailand (to TA and NP)].

Response: We have removed the funding statement from our acknowledgments section and included the funding statement above in the cover letter as requested.

Comment 6: Thank you for uploading your study's underlying data set. Unfortunately, the repository you have noted in your Data Availability statement does not qualify as an acceptable data repository according to PLOS's standards.

At this time, please upload the minimal data set necessary to replicate your study's findings to a stable, public repository (such as figshare or Dryad) and provide us with the relevant URLs, DOIs, or accession numbers that may be used to access these data.

Response: We have uploaded our data to FigShare and included a link to the repository in our methods section (line 700). The link is: https://figshare.com/s/e1a6fa2ff4a5e815808b

Comment 7: Please upload a copy of Supporting Information FigureTable S1 to S6 which you refer to in your text on page 21.

Response: Instead of making this available as supporting information, we now provide a link to the figshare repository (https://figshare.com/s/e1a6fa2ff4a5e815808b). This way the data is more easily discoverable. We indicate where readers can find this information in the methods section (line 769-770). Our supporting information is now limited to one figure and one table, both of which have been uploaded to the PlosOne website.

Comment 8: If the reviewer comments include a recommendation to cite specific previously published works, please review and evaluate these publications to determine whether they are relevant and should be cited. There is no requirement to cite these works unless the editor has indicated otherwise.

Response: We thank you for your concern, our reviewers do not recommend citing specific previously published works.

Reviewer1:

Comment 1: L73 and, and although – delete one and

Response: Thank you for catching this; we have deleted one ‘and’.

Comment 2: L120 I would suggest that Figure 1 should be cited only in Methods section.

Response: Thank you for the suggestion. However, we decided to keep Figure 1 in the results section. Given that the formulation of Jasmin can vary between preparations, we feel it’s important to include this information up front.

Comment 3: L125 Citations to literature in regards to different effects should be moved and/or kept only in discussion section.

Response: We moved the citations to literature in regarding the effects of Jasmine to the Discussion section.

Comment 4: Introduction should contain section framing the paradox of both antigenic and anxiolytic effects.

Response: We have now modified the introduction to better frame the paradoxical drug effects of Arabian Jasmine and how our study is attempting to address this (lines 85-87).

Comment 5: Clarify the ratio of selection of three used strains (AB, TL and WIK).

Response: We are not entirely sure what the reviewer means by the ‘ratio of selection’ of the three strains. We chose these strains because they are amongst the most commonly used in the zebrafish community and they are readily available from reputable sources (e.g., the Zebrafish International Resource Center, ZIRC). We have now added this justification on lines 604-606.

Comment 6: Please use confidence intervals to support interpretation of the near-significant findings in regards of the effects.

Response: We have now added confidence intervals where appropriate to support the interpretation of the near-significant findings.

Reviewer 2:

Comment 1: Pg 11, line 91. To correct: Zebrafish [Danio rerio, Hamilton (1922)]

Response: We have now corrected this (line 90)

Comment 2: Pg 11, lines 101 - 106. Remove this snippet: "Here, we find...". Al snippet is results/conclusion. Delete from introduction.

Response: We have removed this from the introduction.

Comment 3: Rewrite the introduction with updated citations, i.e., from the last 5 years.

Response: We appreciate the suggestion. Most of the studies in the introduction are from the last 5-10 years. However, there is a dearth of relevant studies on Arabian Jasmine and the related phytochemicals (like benzaldehyde and linalool). Thus, the only references available are, unfortunately, quite old. Perhaps our work will spur others to more thoroughly investigate the constituents of Jasmine.

Comment 4: Pg. 12, lines 120-121: 120. Rewrite the snippet: "Arabian jasmine (Fig. 1A) was extracted via ultrasonic-assisted extraction with air pressure (Fig. 1B), using water as the solvent". The way it is written is method. Write in the form of result!

Response: We have now re-written the beginning of the results section and it reads much better (now lines 116-119). Thank you for the suggestion.

Comment 5: Put the "p" of statistical analysis in all text in lowercase

Response: We have made the requested change.

Comment 6: Pg 21, lines 433-434. To correct: [e.g., serotonin transporter (5-HTT) gene]

Response: We have now updated this (now lines 498-500)

Comment 7: Increase the size of figures

Response: We checked our figures and found that they did not quite meet the figure guidelines for font size at PLoS One. To fix this, we separated our figures so they could accommodate larger sizes. Now figure 3 has been broken into two figures.

Comment 8: In the discussion explain the reason for preparing feed with gelatin

Response: We have now added this to the discussion (lines 432-441)

---

## [Editor Report · Decision Letter 1]

3 Sep 2025

The effects of Arabian jasmine on zebrafish behavior depends on strain, sex, and personality

PONE-D-25-26581R1

Dear Dr. Kenney,

We’re pleased to inform you that your manuscript has been judged scientifically suitable for publication and will be formally accepted for publication once it meets all outstanding technical requirements.

Kind regards,

Branislav T. Šiler, Ph.D.

Academic Editor

PLOS ONE
---

## [Editor Report · Acceptance letter]

PONE-D-25-26581R1

PLOS ONE

Dear Dr. Kenney,

I'm pleased to inform you that your manuscript has been deemed suitable for publication in PLOS ONE. Congratulations! Your manuscript is now being handed over to our production team.

Kind regards,

on behalf of

Dr. Branislav T. Šiler

Academic Editor

PLOS ONE